Dissecting antibody-dependent enhancement modulation by Fc-modified cross-neutralizing human monoclonal antibody

Injampa Subenya 1
Benjathummarak Surachet 2
Keadsanti Sujitra 2
Sootichote Rochanawan 2
Puangmanee Wilarat 3
Yamanaka Atsushi 4
Sasaki Tadahiro 5
Ramasoota Pongrama 2 3
Pitaksajjakul Pannamthip pannamthip.pit@mahidol.ac.th 2 3
1 Faculty of Medicine, King Mongkut’s Institute of Technology, Ladkrabang , Bangkok , Thailand
2 Center of Excellence for Antibody Research, Faculty of Tropical Medicine, Mahidol University , Bangkok , Thailand
3 Department of Social and Environmental Medicine, Faculty of Tropical Medicine, Mahidol University , Bangkok , Thailand
4 Research Institute for Microbial Diseases, Osaka University , Osaka , Japan
5 Department of Viral Infections, Research Institute for Microbial Diseases, Osaka University , Osaka , Japan
Alpuche Juan
Electronic publication date: 2025 Nov 19
Publication date: 2025
Volume: 13
Electronic Location ID: e20329
Received 2025 Jun 3; Accepted 2025 Oct 13
Copyright: ©2025 Injampa et al.
Copyright year: 2025
Copyright holder: Injampa et al.
License: This is an open access article distributed under the terms of the Creative Commons Attribution License, which permits unrestricted use, distribution, reproduction and adaptation in any medium and for any purpose provided that it is properly attributed. For attribution, the original author(s), title, publication source (PeerJ) and either DOI or URL of the article must be cited.
License URL: https://creativecommons.org/licenses/by/4.0/

Keywords: Dengue virus, Human monoclonal antibody, Cross-neutralization, Fc-modified

Funding: Center of Excellence in Medical Biotechnology No. SB-61-004-02 SATREPS This work was supported by Center of Excellence in Medical Biotechnology (No. SB-61-004-02), and SATREPS. The funders had no role in study design, data collection and analysis, decision to publish, or preparation of the manuscript.

==============================
Background

Dengue is a mosquito-borne disease caused by four dengue virus serotypes (DENV1 to DENV4). Secondary infections can generate flavivirus cross-reactive antibodies at sub-neutralizing levels. This phenomenon can significantly increase the severity of secondary infections through antibody-dependent enhancement (ADE). ADE is associated with a high risk of viral infection in immune effector cells, triggering cytokine cascades and activating the complement system, which can lead to severe symptoms. Despite extensive studies, therapeutic antibodies, particularly fully human monoclonal antibodies, which could serve as candidates for immune passive therapy, have not yet been discovered.

Methodology

This study generated LALA-mutated human monoclonal antibody clone B3B9 (LALA-B3B9 HuMAb) which can neutralize all four DENV serotypes without enhancing viral activity. The number of infected cells in the ADE assay was compared among the wild-type antibody (B3B9), LALA-B3B9 HuMAb, and an Fc modified variant at position N297Q (N297Q-B3B9), with or without complement proteins. Moreover, the therapeutic efficacy of these HuMAbs against ADE infection by competing with natural antibodies in patients with acute dengue was evaluated using the in vitro suppression-of-enhancement assay in K562 cells.

Result

Our novel Fc-modified antibody LALA-B3B9 (Leu234Ala/Leu235Ala mutations), exhibited neutralizing activity against all dengue virus serotypes without triggering ADE activity at any antibody concentration. This outcome was similar to that observed with the previously developed Fc-modified N297Q-B3B9 antibody (N297Q mutation). We further evaluated the effect of complement protein on the enhancing and neutralizing activities of our Fc-modified antibodies. The results showed that LALA-B3B9 and N297Q-B3B9 HuMAbs were complement-independent, meaning that the reduced binding between complement protein (C1q) and the Fc portion of the antibody left the neutralizing and enhancing activities unchanged. Additionally, both LALA-B3B9 and N297Q-B3B9 HuMAbs demonstrated the suppression-of-enhancement activity in K562 cells induced by human anti-DENV serum antibodies. Overall, this study highlights the main advantages of our EDII-specific HuMAbs in inhibiting in vitro ADE, indicating that they are promising candidates for future dengue treatment.

Introduction

Dengue is an arthropod-borne viral disease that is endemic in tropical regions and represents a major public health concern. Dengue virus (DENV) is a member of the Flaviviridae family. This virus contains a positive-sense RNA genome that encodes three structural proteins (envelope glycoprotein, nucleocapsid protein, and precursor membrane protein) and seven nonstructural proteins (NS1, NS2A, NS2B, NS3, NS4A, NS4B, and NS5) (Falgout & Markoff, 1995; Shukla et al., 2020). The envelope protein plays an important role in both viral attachment and cell fusion. This protein comprises three structural domains (EDI, EDII, and EDIII).

Dengue, a mosquito-borne viral infection that affects millions of people worldwide each year, is caused by any of the four genetically related but antigenically distinct DENV serotypes (DENV1–4). The symptoms can vary and range from mild or asymptomatic to severe hemorrhage and shock due to the phenomenon called antibody-dependent enhancement (ADE) (Narayan & Tripathi, 2020). ADE occurs when antibodies bind to non-neutralizing virus-immunocomplexes, facilitating virus entry into cells via Fc gamma receptors (FcγR) and promoting virus internalization. This increases viral production and leads to the enhancement of DENV pathogenesis.

Over the past decade, although many researchers have developed various antibodies specific to several target epitopes of the dengue virus, there is still no therapeutic antibody available for treatment. Antibodies that recognize complex epitopes, such as E-dimers or EDE (envelope dimer epitopes), have been reported to exhibit strong neutralization activity; however, most of these antibodies are serotype-specific (Dejnirattisai et al., 2015). In contrast, antibodies that specifically bind to the EDII fusion loop region generally demonstrate low levels of neutralization but possess cross-neutralizing activity. These antibodies are the predominant target during DENV infection and are present in higher amounts during secondary infection (Sanchez-Burgos et al., 2020; Rathore, Sarker & Gupta, 2019; Wahala et al., 2009; Lai et al., 2008; Crill & Chang, 2004). To enhance prevention, diagnosis, and treatment strategies for dengue fever, it is essential to develop antibodies that provide strong neutralization across all dengue virus serotypes. However, cross-neutralizing antibodies may carry a higher risk of inducing severe disease through antibody-dependent enhancement (ADE), which is mediated by the Fc region. Therefore, further development is needed to engineer these antibodies to retain broad neutralization while minimizing ADE risk.

The original B3B9 antibody was generated from PBMCs obtained from patients in the acute phase of dengue infection. This antibody neutralized all DENV laboratory serotypes and exhibited stronger neutralizing activity against clinical isolates. Furthermore, B3B9 protected suckling mice from death following DENV challenge (Sasaki et al., 2013). Nevertheless, this HuMAb was observed to exhibit viral enhancement activity in vitro. Epitope mapping revealed that the B3B9 antibody targets residues 107–111 of the conserved N-terminal fusion loop peptide of the E protein domain II which is a major target for human antibodies for cross-neutralizing and enhancing activity (Costin et al., 2013; Deng et al., 2011).

Modification or removal of an antibody’s Fc region can reduce potent receptor signaling and influence its therapeutic performance. Genetic engineering at the glycosylation site at asparagine 297 of the Fc region, such as N297A, N297Q, and N297G substitutions, has been reported to reduce antibody binding to Fc receptors (Leabman et al., 2013; Tao & Morrison, 1989; Balsitis et al., 2010). Another substitution used to reduce IgG binding to Fc receptors is the L234A/L235A (LALA) variant, which is fully glycosylated (Beltramello et al., 2010; Wilkinson et al., 2021). This also showed a marked reduction in Fc–Fc gamma receptor binding (Balsitis et al., 2010; Abdeldaim & Schindowski, 2023; Golay, Andrea & Cattaneo, 2022).

Fortunately, to prevent the binding of the Fc region to Fc gamma receptors, which triggers a range of downstream effector functions that play a key role in mediating ADE in dengue pathogenesis, the modified version of our original HuMAb (N297-B3B9) exhibits cross-neutralization properties and eliminates ADE activity in all DENV serotypes with NT50, ranging from 12 to 0.125 µg/ml (Injampa et al., 2017). This cross-neutralizing HuMAb was generated by mutating the human monoclonal antibody D23-1B3B9 (B3B9) (Setthapramote et al., 2012) at Asn297 of the Fc region of the heavy chain (HC) constant domain 2.

However, changes in the glycosylation pattern of antibodies can affect their biological activity (Irvine & Alter, 2020). Previous studies have reported that mutations at Leu234Ala/Leu235Ala and alterations in Fc glycosylation result in a significant loss of binding affinity to Fc gamma receptors, as well as a notable reduction in binding to C1q and Fc receptors (Wilkinson et al., 2021; Furuyama et al., 2020; Arduin et al., 2015). Therefore, in this study, we produced an alternative (LALA-B3B9) cross-reactive human monoclonal antibody against all DENV serotypes and studied the activities of two versions of the antibody (N297-B3B9 and LALA-B3B9) in relation to complement. Site-directed mutagenesis method was used to generate the Fc modification recombinant human monoclonal antibody by double mutation at positions Leu234Ala and Leu235Ala (commonly referred to as LALA mutation), which are residues in the antibody’s lower hinge region, to the heavy chain construct (LALA-B3B9). This modified heavy chain plasmid was co-transfected with light-chain plasmid construct to HEK 293T mammalian cells to produce full IgG.

When comparing the two versions of Fc-modified antibodies to the wild-type antibody, it is essential to clarify their neutralizing activity and any changes in Fc-mediated immune effector functions resulting from the Fc modifications. This modification in Fc receptor binding may affect activity related to the complement system, which plays a crucial role in reducing viral infection. As evidenced by Yamanaka & Konishi (2017) and Yamanaka, Kosugi & Konishi (2008) the two distinct activities of antibodies (neutralizing and enhancing) are controlled by the level of complement (Mehlhop et al., 2007; Yamanaka & Konishi, 2016). Multiple potential mechanisms by which C1q reduces ADE include steric interference with FcγR-Fc interactions, attenuation of cell signaling through FcγR leading to internalization, restriction of E protein movements required for fusion, and downstream complement-mediated effects, such as membrane attack complex (MAC)-dependent lysis (Mehlhop et al., 2007; Mehlhop et al., 2009). Hence, the complement-dependent antibody activity was assessed to determine whether this modified antibody, in combination with the complement system, influences neutralizing and enhancing activity, which should be characterized as essential information for drug and vaccine development.

Moreover, to simulate the use of these antibodies in dengue-infected patients, this study also examined the effect of EDII specific Fc-modified variants (N297Q-B3B9 and LALA-B3B9) in suppressing ADE in patient serum during the ADE phase in vitro.

Materials & Methods

Virus and cells

The DENV1 Mochizuki strain, DENV2 New Guinea C (NGC) strain, DENV3 H87 strain, and DENV4 H241 strain, kindly provided by Prof. Kazuyoshi Ikuta, Research Institute for Microbial Diseases (RIMD), Osaka University, were propagated in C6/36 cells cultured in Leibovitz L15 medium (Hyclone) with 10% fetal bovine serum (FBS) (Hyclone) and 0.3% tryptose phosphate broth. Vero cells were obtained from Prof. Kazuyoshi Ikuta, RIMD, and used for the neutralization activity test. The cells were maintained in the minimal essential medium (MEM) (GE Healthcare UK Ltd., Buckinghamshire, UK) with 10% FBS. For the ADE test, K562 cells were kindly provided by Assoc. Prof. Atsushi Yamanaka, RIMD, and cultured in the RPMI 1640 medium (Hyclone) supplemented with 10% FBS (Hyclone®, USA). CHO-K1 cells were obtained from Assoc. Prof. Atsushi Yamanaka, RIMD, were used for the stable expression of B3B9, N297Q-B3B9, and LALA-B3B9 antibodies. These cells were cultured in the MEM medium supplemented with 10% FBS and 1% nonessential amino acids (Gibco, Thermo Fisher Scientific, Waltham, MA, USA). Portions of this text were previously published as part of a preprint (https://www.biorxiv.org/content/10.1101/2023.05.25.542225v1.full).

Generation of LALA-mutated human monoclonal antibody clone B3B9

Plasmid construction

The variable genes of the heavy (VH) and light (VL) chains of the B3B9 HuMAb were amplified separately by polymerase chain reaction (PCR) from hybridoma cells (Degorce et al., 2009), and cloned into a plasmid containing the antibody constant region of pQCX plasmid backbone (Injampa et al., 2017; Pitaksajjakul et al., 2014). Antibody mutations were performed over the Fc portion of the heavy chain plasmid regions at positions 234 and 235 from leucine (L) to alanine (A) (L234A, L235A (LALA)) by site-directed mutagenesis with the In-Fusion Cloning System (In-Fusion® HD Cloning Plus; Clontech Laboratories Inc., Shiga, Japan) based on the manufacturer’s protocol (Injampa et al., 2017).

Transient expression of LALA-B3B9 mutation antibody

After sequencing of the mutated heavy chain plasmid for confirmation, the heavy- and light-chain plasmids were transfected to 6.8 × 106 HEK293T cells in a T75 cell culture flask for transient expression. The culture medium was collected and used in immunofluorescence assays to determine the DENV-binding activity. The culture medium containing the secreted LALA-B3B9 recombinant immunoglobulin G (rIgG) was purified using the protein A affinity column (GE Healthcare, Chicago, IL, USA). The eluted fractions from purification were subjected to sodium dodecyl sulfate-polyacrylamide gel electrophoresis to determine the purity of the purified antibody. The LALA-B3B9 rIgG concentration was measured using the BCA protein assay kit (Thermo Scientific, Waltham, MA, USA).

Indirect immunofluorescence assay for the DENV-binding activity

Indirect immunofluorescence assay was used to assess the binding activity of the LALA-B3B9 antibodies for all DENV serotypes. Vero cells were mock-infected or infected with DENV at a multiplicity of infection (MOI) of 0.1 in 96-well cell culture plates for 72 h. After fixing the cells with 3.7% formaldehyde in phosphate-buffered saline (PBS) and permeabilized with 0.1% Triton X-100 in PBS, the cells were stained with culture supernatant from transfected cells as primary antibody. After incubation at 37 °C for 1 h, Alexa Fluor 488-conjugated anti-human IgG (Thermo Fisher Scientific. Invitrogen. Waltham, MA, USA.) (1:1,000) was added as a secondary antibody. The fluorescence can be visualized using a fluorescence microscope (IX71; Olympus, Tokyo, Japan).

Cellular binding activity of B3B9, N297Q-B3B9 and LALA-B3B9 mAbs to CD32a (FcγRIIa, H131)

The Tag-lite binding assay was used to evaluate the binding activity of the Fc domain of the B3B9, N297Q-B3B9, and LALA-B3B9 monoclonal antibodies with FcγRIIa (CD32a) (Cisbio, Codolet, France). In this assay, binding activity of unlabeled antibodies to the receptor were determined by competing with an acceptor labeled human IgG (IgG-d2). HEK293 cells expressing the FcγIIA receptor labeled with Lumi4-terbium Cryptate (Lumi4-Tb) were dispensed, followed by the addition five µl of IgG1 labeled with d2 and five µl of the unlabeled sample (B3B9, N297Q-B3B9, or LALA-B3B9 mAb) in a concentration ranging from 3–5 nM. After 2 h of incubation at room temperature, the signal was measured at 620 nm for the fluorescent of Lumi4-Tb and at 665 nm for the fluorescent acceptor dye (d2). A standard curve was generated from a known concentration of IgG1 Protein (Degorce et al., 2009). Data obtained were compared with the serum-free media used as a negative control and human IgG1 protein (D239E, L241M, HEK293) (MedChemExpress), which is the reference recombinant human-derived IgG1 protein expressed by HEK293 and interacts with the Fc gamma receptor IIa, used as the positive control.

Foci reduction neutralization test of LALA-B3B9 antibodies

The neutralization activity of the LALA-B3B9 antibodies was assessed and compared with that of the previously generated aglycosylated HuMAb (N297Q-B3B9) (Injampa et al., 2017) and wild-type antibody (B3B9) using Vero cells, as described in a previous study (Pitaksajjakul et al., 2014). In brief, DENV at an MOI of 0.01 was preincubated with the respective purified LALA-B3B9 or N297Q-B3B9 or B3B9 HuMAb at 37 °C for 1 h before infecting Vero cells in 96-well-cell culture plates. At 2 h post-infection, the plate was overlaid with 2% carboxymethyl cellulose in the MEM medium with 2% FBS. The cells were incubated for 2 days for DENV4 and for 3 days for DENV1, DENV2, and DENV3. Then, the cells were fixed and immune-stained using anti-DENV human antibody, followed by Alexa-conjugated anti-human IgG (H+L) (1:1,000 dilution). Neutralization activity was quantified by counting the number of foci for each antibody concentration under a fluorescence microscope compared with the negative control (absence of antibody). The neutralizing antibody titer was considered the minimum IgG concentration yielding a 50% reduction in the focus number (FRNT50). In this study, the anti-influenza virus HuMAb (5E4) was used as an unrelated control.

Antibody-dependent enhancement assay of LALA-B3B9 antibody

The ADE activity of the LALA-B3B9 antibody was assessed using semi-adherent FcγRIIa-bearing K562 cells and compared with that of N297Q-B3B9 and B3B9 (Konishi, Tabuchi & Yamanaka, 2010). In total, 36 microliters of serially diluted antibodies were mixed with 50 µl of DENV in 10% FBS RPMI medium in 96-well poly-L-lysine-coated plates (Corning Inc., NY, USA). The mixture was incubated for 2 h at 37 °C, supplied with 5% CO2. The mixture was then added with 50 µl K562 cells at a density of 2 × 106 cells/ml and incubated at 37 °C under 5% CO2 for 2 days. Thereafter, the cells were washed with PBS three times, dried, and fixed with an acetone/methanol fixing solution at −20 °C for at least 30 mins, and immune-stained with D23–3E6D7, an in-house anti-DENV human antibody specific to E protein (Setthapramote et al., 2012), followed by Horseradish peroxidase-conjugated anti-human IgG (H+L) (Sigma-Aldrich, St. Louis, MO, USA). The signal was developed with a DAB substrate solution (KPL, Gaithersburg, MD, USA). The ADE activity was measured by counting virally infected cells manually under a light microscope at a magnification of 10x. Stained cells were counted in three random fields using a 10 × 10 GRID Microscope Eyepiece Micrometer. The total infected cell count in a well was calculated from the average number of stained cells in one grid multiplied by 153.86 (the area of one well was 153.86 times greater than grid square). The cut-off value for distinguishing enhancing from non-enhancing activities was calculated from the percentage of infected cells obtained using six negative controls. This value represents the number of infected cells when the virus was added to cells without antibody, tested in triplicate assays within each ADE experiment, with the experiments performed in two independent replicates. The mean percentage of infected cells from these negative controls was used to calculate the cut-off value, defined as the mean ± one standard deviation (mean ± SD). Antibody concentrations resulting in a higher or lower number of infected cells compared with the cut-off value were considered indicative of ADE or neutralization, respectively.

Stable expression of B3B9, N297Q-B3B9, and LALA-B3B9 antibodies

The generation of a stable antibody-secreting cell line was described in a previous study (Injampa et al., 2017). Briefly, CHO-K1 cells were seeded at 1.3 × 106 cells in a six-well-cell culture plate. The constructed plasmids expressing the heavy and light-chains, which contained puromycin- and hygromycin-resistant genes, respectively, were co-transfected into the cells using transfection reagent (Lipofectamine® 2000; Thermo Fisher Scientific, Waltham, MA, USA). After 2 days, the culture medium was replaced with MEM supplemented with 10% FBS and 1% NEAA containing puromycin and hygromycin at eight and 800 µg/ml, respectively. After 7 days, when all cells in the control well without plasmid had died, the cells that survived under antibiotic selection were expanded and stored as stable pool. Some cells were used for cloning by limiting dilution in 96 well-cell culture plates with selection media. The selected single-colony clones that tested positive for dengue virus binding by IFA were scaled up for subsequent antibody preparation and characterization.

Complement-induced antibody-dependent activity assay

Serial antibody dilutions were incubated with DENV with or without rabbit complement (Cedarlane, Ontario, Canada) for 2 h at 37 °C. The mixture was mixed with 50 µl of 2 × 106 K562 cells/ml and incubated for 2 more days, followed by fixation and immunostaining of the cells. To differentiate enhancing from neutralizing activities, the cut-off value was determined using the percentage of infected cells measured in six negative controls. These controls—where virus was added to cells without antibody—were tested in triplicate within each assay, and the entire experiment was conducted in two independent replicates. The cut-off value was calculated as the mean percentage of infected cells from the mean ± one standard deviation (mean ± SD). Antibody concentrations resulting in infected cell percentages above or below this cut-off were interpreted as indicative of ADE or neutralization, respectively (Yamanaka, Kosugi & Konishi, 2008).

In vitro suppression-of-enhancement assay in K562 cells

Serum preparation

A single DENV2 human serum sample collected from patients with dengue infection at the acute phase was used in this assay (Anasir & Poh, 2022). After collecting whole blood samples, the serum was separated via centrifugation, heat-inactivated, and frozen at −80 °C. The research protocols for human samples were approved by the Ethics Committee of the Faculty of Tropical Medicine (FTM), Mahidol University (MU) (protocol number FTM ECF-019-05). Informed consent was obtained from the patients before enrollment.

In vitro suppression-of-enhancement assay

First, the optimal serum dilution for DENV2 infected patient showing the greatest enhancement of distinct dengue virus serotypes in K562 cells was determined (1/4,000 serum dilution for DENV1 and DENV2, 1/1,000 serum dilution for DENV3 and DENV4). The patient serum at the predetermined concentration was then mixed with DENV at an MOI of 0.1 and incubated for 1 h at 37 ∘C with 5% CO2 prior to the addition of two-fold serially diluted HuMAbs starting at 1,000 μg/ml. After incubation for 1 h at 37 °C, 50 µl of 2 × 106 cells/ml K562 cells were added to the serum–virus–antibody mixture. At 48 h post-infection at 37 °C with 5% CO2, the cells were fixed by acetone/methanol fixing solution at −20 °C, and immunostaining for the viral antigen was performed by incubating with D23–3E6D7, an in-house anti-DENV human antibody (Setthapramote et al., 2012) for overnight at 4 °C. After incubation, the plate was washed three times and incubated with Horseradish peroxidase-conjugated antihuman IgG (H+L) (Sigma-Aldrich, St. Louis, MO, USA) diluted in 0.05% Tween-20 and 1% FBS in PBS for 1 h at 37 °C. To visualize the infected cells, the signal was developed with a DAB substrate solution (KPL, Gaithersburg, MD, USA). The cut-off value to distinguish enhancing from neutralizing activities was based on the percentage of infected cells in six negative controls, where virus was added without antibody. These were tested in triplicate across two independent experiments. The cut-off was set as the mean ± one standard deviation (mean ± SD) (Williams et al., 2013).

Statistical analysis

All results were expressed as mean ± SD. All calculations were performed using the GraphPad Prism 6 software.

Ethical statement

All experimental procedures using human samples were preapproved by the Ethics Committee of the Faculty of Tropical Medicine (FTM), Mahidol University (protocol number: fTM ECF-019-05). All donors provided a written informed consent before enrollment.

Results

Generation and cross-reactivity of the Fc-modified LALA-B3B9 mutation antibody

The substitution of amino acids at positions 234 and 235 from leucine to alanine in the Fc portion of monoclonal antibody B3B9 was confirmed by DNA sequencing. The cross-reactivity of the LALA-B3B9 antibodies to all DENV serotypes was determined using an immunofluorescence assay and compared with a negative control of mock-infected cells. Results showed that LALA-B3B9 antibodies reacted with all DENV serotypes (Fig. 1A). The purity of the produced LALA-B3B9 antibody was assessed by western blot analysis, which detected a specific IgG band at approximately 250 kDa under non-reducing conditions (Fig. 1B).

Figure 1 Binding activity and expression analysis of the LALA-B3B9 antibody.

(A) Indirect immunofluorescence assay presented the reactivity of culture supernatants of HEK293T cells transiently expressing target antibody reacted to the envelope protein of four DENV serotypes, with negative control of mock infected cells. (B) Western blot analysis of human IgG from purified culture fluid of LALA-B3B9 HuMAb demonstrated the expected band size of full-length IgG. This experiment was conducted under non-reducing conditions, with Human IgG (31154; Thermo Fisher Scientific) as a loading control.

Cellular binding activity of B3B9, N297Q-B3B9 and LALA-B3B9 mAbs to CD32a (FcγRIIa, H131)

The binding activity of the Fc portion of the generated antibodies with the Fc receptor was measured using a competition assay format, utilizing the cell surface bound FcγIIa receptor and the antibody variant. The result in Fig. 2. revealed that the B3B9 antibody effectively competes with acceptor-labeled human IgG for binding with FcγRIIa, resulting in a decrease fluorescence signal at every antibody dilution compared to the negative control (serum-free medium). In contrast, LALA-B3B9 and N297Q-B3B9 demonstrated similar binding behavior, with no significant difference in the fluorescent signal compared to the negative control. Furthermore, these antibodies exhibited significantly lower binding capability to the Fc portion than the positive control protein (Human IgG1). These results suggest that the two Fc modified antibodies diminished the binding between their antibody Fc and FcγRIIa.

Figure 2 Cellular binding activity of B3B9, N297Q-B3B9 and LALA-B3B9 mAbs to FcγRIIa.

Cellular binding activity was determined by a competitive binding assay to assess IgG binding to membrane bound FcγRIIa on HEK cells. Human IgG1 with a label suitable for detection in the TagLite Fluorescence resonance energy transfer (FRET) assay was added at a constant concentration and the different antibody concentrations were added in the indicated concentrations for competing with the Human IgG1 with a label. HEK cells mixed with Human IgG1 served as the positive control, while HEK cells mixed with serum-free media were used as the negative control. The decreasing signal intensity means absence of FRET due to an outcompeting antibody. The results presented as the mean ± SD of duplicate independent experiments in different time points. Statistical significance was analyzed using Kruskal–Wallis analysis. * P < 0.05 was considered significant.

Neutralization of all dengue virus serotypes by the Fc-modified LALA-B3B9, N297Q-B3B9 and B3B9 antibody

The FRNT50 test was used to assess the capabilities of LALA-B3B9 HuMAbs to neutralize DENV1–4 (Arduin et al., 2015). The neutralizing abilities of the HuMAbs were compared among B3B9, wild-type antibody, and N297Q-B3B9 variant, an Fc-modified antibody with a substitution at position N297Q, as described in Fig. 3. All HuMAbs were cross-reactive with four serotypes of dengue virus with different neutralizing potency. At the highest concentration of 64 μg/ml, all HuMAbs reduced DENV2 and DENV4 by 100% and DENV3 by nearly 90%. For DENV1, LALA-B3B9, N297Q-B3B9, and B3B9 antibodies did not achieve 100% neutralization. The FRNT50 values of the three generated antibodies are shown in Table 1. These values differed significantly for dengue serotypes 1, 3, and 4 (p < 0.05), indicating significant differences in neutralization potency among the antibodies. In contrast, no significant differences were observed among the antibodies for dengue serotype 2 (p > 0.05). In terms of neutralization potency, the LALA-B3B9 antibody exhibits moderate activity against DENV2 and DENV4, with 50% neutralization titers ranging from 0.1 to 1 µg/ml., with FRNT50 values at 0.79 (±0.27) and 0.8 (±0.265) µg/ml, respectively. For DENV3, the FRNT50 concentrations of the LALA-B3B9 antibody was 4.97 (±0.535) µg/ml. The FRNT50 concentrations of N297Q-B3B9 against DENV1 and DENV3 were lower than those of LALA-B3B9 and B3B9 HuMAb but higher against DENV4 (p < 0.05).

Figure 3 Neutralization activity of LALA-B3B9 and N297Q-B3B9 HuMAbs.

Neutralizing activity against DENVs using the average of two independent experiments is shown. Dotted lines indicate 50% neutralizing activity. The error bars show standard deviation of the percent reduction of each antibody concentrations obtained from two experiments.

Table 1 The FRNT50 values compared between LALA-B3B9, N297Q-B3B9 and B3B9 antibodies against all serotypes of dengue virus.

All antibodies exhibited significant differences in their FRNT50 concentrations against DENV1 (p < 0.05), but no significant differences against DENV2 (p > 0.05). For DENV3 and DENV4, there was no significant difference in FRNT50 concentrations between LALA-B3B9 and B3B9 antibodies (p < 0.05). Statistical significance was analyzed using Kruskal–Wallis analysis.

Antibody	Neutralizing activity (FRNT50) (µg/ml)	
	DENV1	DENV2	DENV3	DENV4	
LALA-B3B9	24.8 ± 1.3	0.79 ± 0.27	4.97 ± 0.535	0.8 ± 0.265	
N297Q-B3B9	3.12 ± 1.0	0.38 ± 0.135	0.40 ± 0.4	5.49 ± 0.665	
B3B9	43.1 ± 3.3	0.485 ± 0.086	4.635 ± 0.995	0.885 ± 0.146	

ADE activity of the Fc-modified B3B9 LALA mutation antibody

The higher infection rates of immune effector cells occur via the ADE phenomenon, which is mediated by Fc–FcR interactions. Hence, not only neutralizing activity but also ADE activity is a substantial concern for antibody-based therapeutic development. In this study, the ADE activity was determined using FcγRII-bearing K562 cells. Figure 4 shows the comparison between wild-type antibody (B3B9), aglycosylated antibody (N297Q-B3B9), and Fc-modified antibody (LALA-B3B9). The neutralizing and enhancing activities were assessed by comparing the number of infected cells at each antibody concentration to the control without antibody. The results showed that ADE activity was completely abolished for the Fc-modified antibodies N297Q-B3B9 and LALA-B3B9 across all antibody concentrations. At higher concentrations, these variants effectively neutralize all dengue virus serotypes. In contrast, the B3B9 antibody exhibited weak neutralizing activity, particularly against DENV1, and induced ADE at sub-neutralizing concentrations across all serotypes.

Figure 4 ADE activity of HuMAbs.

Neutralization and enhancement activity against the four DENV serotypes in FcγRII-bearing K562 cells. The number of infected cells (log10) from each antibody concentration was used to determine the enhancing activity. There were no differences in the neutralization and enhancement activity of the Fc-modified antibodies (p > 0.05). The B3B9 antibody showed a higher number of infected cells than Fc-modified antibodies for DENV1–4 at antibody concentration ranges of 0.006–400, 0.024–1.56, 0.006–100 and 0.00038–100 µg/ml, respectively. Dotted lines represent the mean number of infected cells of no antibody control ± SD. Each data point indicates the average result obtained from two independent ADE assays. Statistical significance was analyzed using Kruskal–Wallis analysis.

Complement-induced antibody-dependent activity assay

Since complement can play a role in the protective activities against viral infection, Fc modification not only reduces Fc receptor binding but also decreases binding to the C1q complement protein. Therefore, the balance between the enhancing and neutralizing activities of both types of modified antibodies and the wild-type antibody were investigated using the complement-induced antibody-dependent activity assay (Yamanaka & Konishi, 2017). For the wild-type antibody, complement proteins helped reduce ADE activity across all dengue virus serotypes, particularly DENV1 and DENV2, where ADE was significantly abolished. The number of infected cells during ADE activity in the presence of complement was significantly lower than those in the absence of complement, at antibody concentrations of 1.56–25 µg/ml for DENV1, 0.097–1.56 µg/ml for DENV2, 0.39–25 µg/ml for DENV3, and 0.006–400 µg/ml for DENV4 (Fig. 5A). However, despite the presence of complement, enhancing activity was still observed against DENV3 at antibody concentrations higher than 0.024 µg/ml and against DENV4 at all concentrations.

Figure 5 Complement-induced antibody-dependent activity assay of N297Q-B3B9 (A), LALA-B3B9 (B), and B3B9 MAbs (C) against all dengue virus serotypes.

Comparison of the number of infected cells from each antibody concentration with and without complement using FcγRII-bearing K562 cells. Average of two independent experiments is shown in each data point. Dotted lines display cut-off values for differentiating enhancing and neutralizing activity (the mean number of infected cells of no antibody ± SD). The error bars show SD of the two repeated experiments. The significances of differences in number of infected cells were evaluated by the Student’s t test.

In contrast, for the Fc-modified antibodies, there was no significant difference in the neutralization potency of LALA-B3B9 and N297Q-B3B9 HuMAbs in the presence or absence of complement proteins (Figs. 5B, 5C). These antibodies can neutralize all four DENV serotypes without enhancing infectivity. Therefore, both the neutralizing and enhancing activities of N297Q-B3B9 and LALA-B3B9 were complement-independent.

In vitro suppression-of-enhancement assay in K562 cells

The serum samples of patients with acute DENV-2 infection at enhancing concentrations were used to predict the competitive ability of the Fc-modified antibodies and wild-type antibodies to natural dengue infection. The use of human serum samples was approved by the Ethical Committees of the Faculty of Tropical Medicine, Mahidol University. The number of infected cells was compared with that of controls (the mean number of cells infected by a mixture of virus and patient serum without antibody ±SD) to determine the neutralizing or enhancing activity. Each generated antibody dilution was mixed with the DENV2 serum at an enhancing concentration that pre-incubated with virus. Then, K562 cells were added. LALA-B3B9 and N297Q-B3B9 exhibited neutralizing activity against all virus serotypes without enhancing their infectivity. There were no differences in the neutralization and enhancement activity of the two Fc-modified antibodies (p > 0.05).

The B3B9 antibody showed a higher number of infected cells than those from both the no antibody control and Fc-modified antibodies for DENV1 –4, at antibody concentration ranges of 15.625-1000, 1.95–62.5, 1.95-250 and 1.95 –31.25 µg/ml, respectively. This antibody presented with enhancing activity against DENV1 at antibody concentrations of 500–15.63 µg/ml and DENV3 at antibody concentrations of above 31.25 µg/ml. However, it was able to neutralize DENV2 and DENV4 at all antibody dilutions (Fig. 6).

Figure 6 In vitro suppression-of-enhancement assay in K562 cells.

The peak enhancing titer for DENV2-human immune serum was mixed with dengue virus and each type of antibodies (B3B9, LALA-B3B9 and N297Q-B3B9) at different dilutions and infected to K562 cells. The infection rate was expressed as number of infected cells. Average of two independent experiments is shown in each data point. Dotted lines display cut-off values for differentiating enhancing and neutralizing activity (the mean number of infected cells of mixture of virus and patient serum without antibody ± SD). The error bars show SD of the two repeated experiments. Statistical significance was analyzed using Kruskal–Wallis analysis. Probability values (P) of less than 0.05 were considered significant.

Discussion

Currently, various techniques have been used to develop DENV- neutralizing HuMAbs, however no specific therapeutic agent for flavivirus infection is available. We thus established a modified Fc antibody against DENV as another therapeutic candidate, introducing double mutation at position L234A/L235A (LALA). This approach was based on previous studies showing that these substitutions reduce binding between the Fc–Fc receptors (FcγRI, FcγRII, and FcγRIII) (Balsitis et al., 2010; Abdeldaim & Schindowski, 2023; Williams et al., 2013). The development of this antibody resulted from the successful generation of the Fc-modified N297 antibody (N297Q-B3B9). This modified antibody can neutralize all DENV serotypes and eliminate the ADE activity in all serotypes in both FcRI- and RII-bearing THP-1 cells and FcRII-bearing K562 cells (Injampa et al., 2017). This aglycosylated human monoclonal antibody targets the fusion loop of envelope protein domain II, which is the major surface protein containing most neutralizing epitopes and capable of eliciting broadly neutralizing antibodies (Ramanathan et al., 2016; Kotaki et al., 2021; Fibriansah et al., 2021). Previous studies have found that antibodies against E protein of DENV play an important role in both protection and enhancement of the disease, particularly following the primary infection. Moreover, the highly conserved residues at the fusion loop of domain II are the epitopes recognized by the predominantly cross-reactive anti-E antibodies (Lai et al., 2008). This is considered an advantage of this antibody as a result of its target epitope. In particular, EDII contains the fusion loop, which is important for viral–host membrane fusion and cell entry (Anasir & Poh, 2022). Hence, these viral protein components of dengue can be effective targets for the development of cross-reactive antiviral agents.

The binding of the Fc region of each modified antibody with FcγRIIa was determined using a competition assay. The results of binding activity verified that Fc-modified antibodies abrogated the interaction between antibody Fc and FcγRIIa. In contrast, wild-type antibody can compete with FcγRIIa binding protein. The neutralizing and enhancing activity of the Fc variant L234A/L235A antibody was compared with that of the aglycosylated Fc-modified (N297Q-B3B9) and wild type (B3B9) antibody. Both the Fc-modified N297Q-B3B9 and LALA-B3B9 antibodies are cross-reactive against all dengue virus serotypes and show neutralizing activity across different serotypes. Notably, all antibodies completely inhibit infection by dengue virus serotype 2; however, they do not achieve complete neutralization of serotype 1. B3B9 HuMAb and its Fc-modified forms, LALA-B3B9 and N297Q-B3B9, exhibited weak neutralizing activity against Dengue virus serotype 1 (DENV-1), likely due to differences in the antibody’s binding affinity. This weak activity may offer limited protection against DENV-1 infection in vivo, potentially failing to prevent viral entry or replication effectively. However, in our previous study (Sasaki et al., 2013), we found that the B3B9 antibody showed stronger neutralizing activity against viruses isolated from patients in endemic areas compared to laboratory-adapted virus strains. Therefore, the neutralizing activity of the modified antibodies against dengue clinical isolates should be studied further. When comparing the NT50 of Fc-modified antibodies with wild-type antibody, we found that the virus neutralization efficiency of the wild-type antibody was not significantly different from that of the LALA-mutated antibody, except for DENV1. The NT50 of LALA-B3B9 antibody differs from that of N297Q-B3B9 antibody in all serotypes except for DENV2. Therefore, the NT50 values among HuMabs modified at different Fc positions failed to demonstrate a clear relationship between neutralizing potency. Since complement system is one of the antibody-mediated effector functions influenced by the binding of the Fc-Fc receptor and the binding of the Fc portion to the C1q complement protein. This system is crucial for protection against viral infections. Complement 1q (C1q) initiates the complement cascade by binding to the Fc portion of the antibody and the Fc gamma receptor (FcγR) on immune effector cells, leading to the formation of the C5b-C9 membrane attack complex, which can lyse the viral envelope (Byrne & Talarico, 2021). Several studies have shown a protective effect of complement against dengue virus. In vitro and in vivo assays demonstrated that C1q-dependent IgG inhibits ADE activity, and the balancing of neutralization and enhancement activity of antibodies is regulated by complement levels within the physiological range (Yamanaka, Kosugi & Konishi, 2008; Mehlhop et al., 2009; Douradinha et al., 2014). We then further investigated how complement influences the neutralization and enhancing activity of both wild-type and modified antibodies. Our study shows that the neutralizing efficacy of the wild-type antibody (B3B9 HuMab) is unaffected by the presence of rabbit complement. In addition, the enhancing activity was reduced, although not completely eliminated, when complement proteins were applied to some dengue serotypes. However, the presence or absence of the complement protein did not significantly affect the neutralizing or enhancing effects of DENV with N297Q-B3B9 and LALA-B3B9 HuMAbs. These two Fc-modified antibodies, with and without complement proteins had a similar neutralizing activity, and the enhancing activity was inhibited at every antibody concentration. Hence, these two antibodies function as complement-independent neutralizing antibodies in DENV infection. Nevertheless, this study used rabbit complement protein to evaluate the effects on Fc-modified antibody functions. It is important to acknowledge that human C1q may more accurately reflect physiological interactions in humans, particularly when studying Fc-mediated functions. This aspect warrants further investigation in future studies.

To enlighten the application of the modified antibodies during the viremia phase of dengue, acute DENV2-immune serum was used in the enhancement-suppression assay for all dengue virus serotypes for predicting in vivo outcomes (Williams et al., 2013). By adding acute DENV2-immune serum which showed cross-reactivity to all 4 DENV serotypes as an enhancing antibody, the serum concentration with peak enhancement was used in this assay to simulate infection enhancement derived from the non-neutralizing or sub-neutralizing antibodies presented in human serum. Serum from patients in the acute phase, which the predominant antibody population targets the envelope protein similar to that of our generated anti-E human monoclonal antibodies (Thammasonthijarern et al., 2020), was used to better assess competitive binding ability. The results demonstrated that the two Fc-modified antibodies could compete with most serum antibodies for DENV binding or reduce the enhancing activity of natural antibodies, thereby eliminating serum-derived enhancing activity. We observed differences in the ability of the wild-type antibody to prevent antibody-enhanced DENV infection when DENV-immune human serum was used. This wild-type antibody could compete with serum antibodies for infection with DENV2 and DENV4, but not for DENV 1 and 3. The data suggest that N297Q-B3B9, LALA-B3B9, and B3B9 MAb have a higher affinity than the enhancing antibodies present in patient serum, allowing these antibodies to bind and/or displace the enhancing antibodies. However, this ability cannot be inferred for wild-type antibodies against DENV 1 and 3. These are major advantages of our anti-EDII antibodies compared with antibodies targeted to other regions, even with anti-EDIII (Wahala et al., 2009) or EDE antibodies (Williams et al., 2013; Smith et al., 2014). This may be due to the fact that this antibody was selected from several hundred hybridoma clones derived from antibody-producing cells in Thai patients infected with dengue (Setthapramote et al., 2012). The ability to predict the activity of these Fc-modified HuMAbs in the serum provides novel insights into the mechanism by which modified human monoclonal antibodies can prevent antibody enhancement by competing with natural antibodies in the serum of patients, while the natural antibodies do not alter the neutralizing and enhancing activity of these generated antibodies. Based on our results, the variants of Fc-modified antibodies are considered more advantageous for dengue treatment.

Overall, the novel Fc-modified human monoclonal antibody LALA-B3B9 can neutralize all DENV serotypes without enhancing its activity. This property is similar to that of the previously generated aglycosylated antibody (N297Q-B3B9). Moreover, these two Fc-modified HuMAbs can compete with enhancing antibodies to reduce viral enhancement. Considering these promising results, further studies on immune effector functions mediated by Fc–Fc interaction, such as targeted cell elimination via antibody-dependent cellular cytotoxicity and phagocytosis, should be conducted to determine the efficacy of the modified HuMAbs in mediating protection, as well as the studies using DENV clinical isolates. We acknowledge, however, that our results are derived from assays in K562 cells, which may not fully represent primary monocytes, the physiologically relevant target cells for ADE. Therefore, validating these findings in primary monocytes and in vivo models that account for the role of neonatal Fc receptors will be essential to demonstrate and further substantiate the efficacy of these Fc-modified antibodies. In addition, a previous study found that N-glycans are essential not only for immune effector functions but also for stability under physiological or low-temperature conditions, pharmacokinetics, and biodistribution (Abdeldaim & Schindowski, 2023; Alsenaidy et al., 2013; Higel et al., 2016). To determine the potential drawbacks of aglycosylated antibodies, the biological activity and stability of these Fc-modified antibodies need to be further studied.

Conclusions

This study generated LALA-B3B9, an Fc-modified antibody targeting the fusion loop on EDII (Injampa et al., 2017). This antibody has several features that make it appropriate for potential therapeutic applications: it can neutralize all DENV serotypes without enhancing viral infectivity. Moreover, this study found that LALA-B3B9 and the previously generated Fc-modified antibody, N297Q-B3B9, function as complement-independent neutralizing antibodies, indicating that the modification does not affect their protective efficacy. Moreover, this study emphasized the benefit of anti-EDII Fc-modified antibodies in the suppression of serum derived infection enhancement.

Based on the reasons mentioned above, the effect of the LALA-mutated antibody is similar to that of a previously generated aglycosylated (N297Q-B3B9). Therefore, this could be a benefit of this antibody, as it does not cause changes in biological activity due to alterations in N-glycosylation. However, changes in Fc-Fc receptor binding can affect the antibody’s ability to perform other functions, such as antibody-dependent cellular phagocytosis (ADCP) or Antibody-dependent cellular cytotoxicity (ADCC), which should be further studied before development for future use. This study offers important insights into neutralization and ADE mechanisms using K562 cells, an established model that expresses FcγRII and has been widely applied in ADE research. While these cells provide a useful platform for investigation, future studies using primary human monocytes—the cell type most relevant to ADE in vivo—could offer an even more complete understanding of the magnitude and mechanisms involved. In addition, exploring the role of neonatal Fc receptors (FcRn), which may influence antibody half-life and distribution in vivo, could further enrich our knowledge. Together, such approaches will help build on the present findings and deepen our understanding of ADE in physiological contexts.

Supplemental Information

Supplemental Information 1 Cellular binding activity of B3B9, N297Q-B3B9 and LALA-B3B9 mAbs to Fc g RIIa

Supplemental Information 2 Neutralization activity of LALA-B3B9 and N297Q-B3B9 HuMAbs

Supplemental Information 3 Raw data for ADE titer in in vitro suppression-of-enhancement assay in K562 cells

Supplemental Information 4 ADE activity of HuMAbs

Supplemental Information 5 Complement-induced antibody-dependent activity assay of N297Q-B3B9 (A), LALA-B3B9 (B), and B3B9 MAbs (C) against all dengue virus serotypes

Supplemental Information 6 In vitro suppression-of-enhancement assay in K562 cells

Supplemental Information 7 The enhancement curve for acute DENV-2 serum utilized in the in vitro suppression assay

Serial dilutions of serum from a patient with acute DENV-2 infection were incubated with K562 cells to assess infection rates. The optimal dilutions yielding the greatest enhancement of dengue virus serotypes were 1:4000 for DENV-1 and DENV-2, and 1:1000 for DENV-3 and DENV-4.

The authors would like to thank Dr. Nipa Thummasonticharoen for collecting dengue patient blood samples, and all participants who were enrolled in the study. The authors also thank Prof. Kazuyoshi Ikuta, Research Institute for Microbial Diseases, Osaka University for providing the Vero cell line and dengue virus strains.

Additional Information and Declarations

Competing Interests

Author Contributions

Human Ethics

Data Availability

The authors declare there are no competing interests.

Subenya Injampa conceived and designed the experiments, performed the experiments, analyzed the data, prepared figures and/or tables, authored or reviewed drafts of the article, and approved the final draft.

Surachet Benjathummarak performed the experiments, prepared figures and/or tables, and approved the final draft.

Sujitra Keadsanti performed the experiments, prepared figures and/or tables, and approved the final draft.

Rochanawan Sootichote performed the experiments, prepared figures and/or tables, and approved the final draft.

Wilarat Puangmanee performed the experiments, prepared figures and/or tables, and approved the final draft.

Atsushi Yamanaka conceived and designed the experiments, authored or reviewed drafts of the article, and approved the final draft.

Tadahiro Sasaki conceived and designed the experiments, authored or reviewed drafts of the article, and approved the final draft.

Pongrama Ramasoota conceived and designed the experiments, authored or reviewed drafts of the article, and approved the final draft.

Pannamthip Pitaksajjakul conceived and designed the experiments, authored or reviewed drafts of the article, and approved the final draft.

The following information was supplied relating to ethical approvals (i.e., approving body and any reference numbers):

All experimental procedures using human samples were preapproved by the Ethics Committee of the Faculty of Tropical Medicine (FTM), Mahidol University (protocol number: FTM ECF-019-05).

The following information was supplied regarding data availability:

The raw data of the analysis are available in supplement excel and figure file.

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
