# Peer review of "Dissecting antibody-dependent enhancement modulation by Fc-modified cross-neutralizing human monoclonal antibody"

_PeerJ, doi:10.7717/peerj.20329_

## Round 0.1 · original submission · Major Revisions

· Academic Editor

Major Revisions

**Language Note:** The review process has identified that the English language must be improved. PeerJ can provide language editing services - please contact us at [email protected] for pricing (be sure to provide your manuscript number and title). Alternatively, you should make your own arrangements to improve the language quality and provide details in your response letter. – PeerJ Staff

Reviewer 1 ·

Basic reporting

The manuscript entitled “Generation and therapeutic efficacy of the Fc-modified cross-neutralizing human monoclonal antibodies against dengue virus” by Injampa et al. describes the generation of the LALA-mutated human monoclonal antibody clone B3B9 (LALA-B3B9 HuMAb) and the in vitro characterization of both neutralizing and antibody-dependent enhancing activities. The study provides valuable insight into the generation and characterization of a novel Fc-modified EDII-specific human monoclonal antibody against dengue virus, which may be a promising candidate for the treatment of dengue infections in the future.

A detailed account of the methodological and analytical procedures is presented in the manuscript.

Some major points to address are listed below:

1) I recommend revising the title, as the study does not assess the therapeutic efficacy of the Fc-modified human monoclonal antibody. Rather, it focuses on evaluating the inhibition of antibody-dependent enhancement (ADE) in vitro, using human anti-DENV2 serum. Additionally, the phrase “therapeutic efficacy” should be eliminated from the manuscript. References should be made to “Fc-modified cross-neutralizing human monoclonal antibody”—omitting both the definite article and any pluralization.

2) In the Abstract (line 50), the term “in vivo ADE” should be corrected to “in vitro ADE”, as the assay utilized human K562 cells, irrespective of the presence of human anti-DENV2 serum.

3) Substantial improvements to the English language in the Introduction and Results sections are necessary to ensure clarity for an international audience. The current phrasing of lines 71-72, 73-75, 77-81, 83-84, 90-91, 96-97, 121-123, 133-135, 298-301, 309-311, 315-316, 317-318, 334-335, 402-403, 425-427, 441-444 makes the comprehension difficult. I suggest rephrasing the indicated lines and editing grammar mistakes throughout the manuscript.

4) In the section detailing the antibody-dependent enhancement assay involving the LALA-B3B9 antibody, it is recommended to specify that semi-adherent K562 cells were employed to clarify the procedure for readers. In line 221, please indicate the origin of the anti-DENV human antibody used, whether it is commercial (brand and country) or obtained from study patients. Also, indicate the brand and country of the commercial antibodies used in all procedures.

5) A clear definition of cut-off values is required across all assays. For instance, lines 229–231 should specify the four negative controls employed in calculating the standard deviation.

6) In the complement-induced antibody-dependent activity assay, why was commercial rabbit complement used instead of commercial human C1q? Given that the enhancing potential of human monoclonal antibodies was assessed using human K562 cells, incorporating human C1q would offer a more appropriate context for analyzing Fc-mediated interactions. This consideration merits further discussion within the study.

7) Why B3B9 HuMAb and the Fc-modified LALA-B3B9 and N297Q-B3B9 Hu MAbs exhibited a weak neutralizing activity against DENV-1? Address the potential implications regarding diminished protective efficacy against DENV-1 infection.

8) In line 434, why are flavivirus infections mentioned? The neutralization and enhancement assays involved exclusively DENV serotypes, without the inclusion of any other flaviviruses.

9) To ensure accurate assessment of ADE inhibition in the presence of modified antibodies, the full enhancement curve for acute DENV-2 serum utilized in the in vitro suppression assay should be provided as Supplementary material.

10) As in vivo therapeutic protection is not assessed, the phrase in line 452 should be omitted.

11) References should be listed in alphabetical order rather than by order of citation within the text.

12) Figure 2, please specify the origin of the labelled human IgG1 used as the positive control.

13) In Figure 5, consider using alternative symbols to enhance the visual distinction between treatment curves.

Some minor points include:

14) In lines 64-65, remove the phrase “ADE activity”.

15) In the Materials and Methods section, avoid using the word “the” before country names and describe all abbreviations.

16) Greek symbols—including mu (μ) for the micro prefix and gamma (γ)—are either omitted or incorrectly rendered in the Materials and Methods and Results sections.

17) In line 268, indicate the brand and country of the anti-DENV E protein human antibody used for immunostaining.

18) In line 311, “cross-reacted” should be replaced with “cross-reactive”.

19) In line 324, “ADE activity” should be replaced with “ADE phenomenon”.

20) In line 356, “acute DENV2” should be replaced with “acute DENV-2 infection”

21) In line 447, remove “lethal” since no antibody-enhanced lethal DENV infection is evaluated in the study.

22) In line 473, “to avoid the drawbacks” should be replaced with “to determine the potential drawbacks”.

23) In line 480, include “potential” therapeutic and remove “their” virus.

24) In lines 481–482, the reference to LALA-B3B9 should be in the singular form, as only one antibody is mentioned.

25) In line 495, “were” enrolled.

Experimental design

-

Validity of the findings

-

Reviewer 2 ·

Basic reporting

The manuscript requires edits to improve the grammar and tenses throughout the manuscript. I would recommend that the authors improve the writing to improve clarity. References are appropriately cited, and results are appropriately presented. In addition, I think the title should be changed to better reflect the study that they are conducting.

Experimental design

Experimental design is appropriate, although the authors did not indicate the number of times the experiments were replicated. Generally, the experiments supported their claims.

Validity of the findings

The in vitro data suggest that the Fc-modified antibodies can mediate neutralisation and not cause ADE. However, all the experiments were performed on K562 cells, which are cancer cell lines and are not derived from human monocytes, which are the cell type susceptible to ADE. I suggest that the authors evaluate the neutralisation and enhancement activity in primary monocytes to validate their findings. Moreover, in vitro studies may not represent the findings in vivo, as the authors did not consider the involvement of neonatal Fc receptors. If the authors do not have the ability to do in vivo studies, they should highlight this important limitation and indicate why their results cannot be extrapolated to what can be seen in vivo.

---

## Round 0.2 · Minor Revisions

· Academic Editor

Minor Revisions

Generalizations beyond K562 cells have caused serious concern among reviewers. Please address this limitation.

Reviewer 1 ·

Basic reporting

All reviewer comments have been addressed, and suggested changes have been introduced in the revised version of the manuscript. These changes have substantially improved the manuscript's clarity. In addition, the authors have introduced the study's limitations.

Experimental design

-

Validity of the findings

-

Reviewer 2 ·

Basic reporting

The authors have acknowledged the limitations, but have not performed additional experiments to demonstrate whether their findings extend beyond K562 cells. If that is the case, then the authors need to include in the discussion section the limitations of their findings and potential future directions related to my concerns.

Experimental design

I have no comments on experimental design.

Validity of the findings

Findings cannot be extended beyond K562. The authors need to highlight the limitations clearly.

---

## Round 0.3 · accepted · Accept

· Academic Editor

Accept

Dear authors, thank you for addresssing all the revierwers´ comments. Now, this manuscript is ready for publication.